# A Two-Step SD/SOCP-GTRS Method for Improved RSS-Based Localization in Wireless Sensor Networks

**DOI:** 10.3390/s25061837

**Published:** 2025-03-15

**Authors:** Shengming Chang, Lincan Li

**Affiliations:** School of Cyber Science and Engineering, Ningbo University of Technology, No. 201, Fenghua Road, Jiangbei District, Ningbo 315211, China; lilincan@nbut.edu.cn

**Keywords:** wireless localization, received signal strength, convex relaxation, weighted least squares, generalized trust region subproblem

## Abstract

Wireless localization is a fundamental component of modern sensor networks, with applications spanning environmental monitoring and smart cities. Ensuring accurate and efficient localization is critical for enhancing network performance and reliability, particularly in the presence of signal attenuation and noise. This study proposes a novel two-step localization framework, SD/SOCP-GTRS, to improve the precision of target localization using received signal strength (RSS) measurements. In the first step (SD/SOCP), semidefinite programming (SDP) and second-order cone programming (SOCP)-based convex relaxation are applied to the maximum likelihood (ML) estimator, generating an initial coarse estimate. The second step (GTRS) refines this estimate using weighted least squares (WLS) and the generalized trust region subproblem (GTRS), mitigating performance degradation caused by relaxation. Monte Carlo simulations validate that the proposed SD/SOCP-GTRS approach effectively reduces root mean square error (RMSE) compared to other methods. These findings demonstrate that the SD/SOCP-GTRS framework consistently outperforms existing techniques, approaching the theoretical performance limit and offering a robust solution for high-precision localization in wireless sensor networks.

## 1. Introduction

Wireless sensor networks (WSNs) have become integral to various applications, such as localization, environmental monitoring, emergency services, precision agriculture, smart homes, and intelligent transportation [1,2,3]. Accurate target localization is crucial for the effectiveness of these networks. In environmental monitoring, for instance, poor localization accuracy may lead to incorrect mapping of pollution sources or wildlife habitats, resulting in flawed ecological assessments and ineffective mitigation strategies [4]. For autonomous systems (e.g., drones or self-driving vehicles), even minor localization errors can cause navigation failures, increasing collision risks and compromising safety [5]. In emergency scenarios, such as disaster response, inaccurate node positioning may delay rescue operations, directly threatening human lives [6]. Furthermore, localization inaccuracies degrade network efficiency by necessitating redundant data retransmissions and excessive energy consumption, ultimately shortening network lifetime [7]. The localization process begins by deploying anchor nodes, whose locations are known, while target nodes estimate their locations using a suitable algorithm. Common localization techniques include time of arrival (TOA) [8,9,10], time difference of arrival (TDOA) [11,12], angle of arrival (AOA) [13,14,15], and received signal strength (RSS) [16,17,18,19,20,21,22], often combined to improve accuracy [23,24,25,26,27,28,29,30,31,32]. Among these, RSS-based localization stands out due to its simplicity, low hardware requirements, and minimal communication overhead. However, RSS measurements are prone to errors caused by log-normal distributions, often approximated by Gaussian distributions.

The problem of RSS-based localization is often tackled using the maximum likelihood (ML) estimator. While the ML estimator is asymptotically optimal, it is nonlinear and nonconvex, and its accuracy is highly dependent on the initial guess. An improper initial guess can result in significant localization errors. To overcome this issue, closed-form solutions and multidimensional scaling (MDS)-based methods have been introduced, which eliminate the need for an initial guess and perform well under low error conditions. A novel approach combining a differential evolution algorithm, opposition-based learning, and adaptive redirection has been proposed. This method avoids approximating the ML cost function, does not require a good initial point, and offers superior performance with relatively low computational complexity in practical scenarios [33].

Convex relaxation techniques, such as semidefinite programming (SDP) and second-order cone programming (SOCP), have been employed to approximate the ML estimator with a convex estimator through relaxation [34,35]. These methods demonstrate robust performance even in noisy environments. Recent studies have increasingly favored convex relaxation methods [19,20,21,23]. In [19], the authors eliminate the logarithmic terms and reformulate the ML problem into a min–max optimization, which is relaxed into an SDP estimator. Similarly, ref. [20] proposes a convex SDP estimator based on relative error estimation. Another approach [21] introduces a new objective function based on least squares (LS) to address the RSS localization problem and uses SOCP relaxation to convert the problem into a convex one.

However, these relaxation-based approaches often suffer from performance degradation due to two key limitations: (1) simplified convex models fail to fully capture the nonlinear effects of signal attenuation and log-normal shadowing, leading to biased estimates in high-noise scenarios [20]; (2) most existing methods neglect the heterogeneous impact of noise across anchor nodes, particularly the stronger influence of distant anchors with weaker RSS signals [19]. Additionally, traditional weighted strategies rely on fixed heuristics (e.g., uniform weights or simple distance thresholds), which cannot adaptively prioritize reliable measurements under dynamic noise conditions [21]. To address these limitations, various nonconvex optimization approaches and weighting strategies have been proposed. For instance, ref. [23] introduces an LS-based nonconvex estimator that approximates the ML estimator and reformulates it within a generalized trust region subproblem (GTRS) framework, effectively reducing computational complexity and improving solution speed. Additionally, ref. [25] proposes an error covariance matrix-based weighting approach to enhance localization accuracy using WLS. Other studies, such as [31,36], introduce distance-based weighting methods to prioritize nearby links for improved accuracy. Similarly, ref. [15] presents a two-step error variance-WLS method, where the first step estimates the target’s location using LS, and the second step refines this estimate with WLS, incorporating weights based on error variance from the first step. This approach mitigates the impact of uncertain anchors and terms, resulting in enhanced accuracy [33]. Despite these advancements, existing studies have not fully addressed the aforementioned gaps, leaving room for further improvement in localization accuracy and network efficiency. By comparing the static nature of traditional weighting strategies with the dynamic nature of the proposed method, it is evident that there is a need for more adaptive and efficient approaches to address the limitations of current techniques.

In this paper, we introduce a two-step localization method, SD/SOCP-GTRS, to achieve globally optimal target localization using RSS measurements. The proposed method directly addresses the above gaps through two synergistic innovations: In Step 1, the convex relaxation process is designed to minimize initial bias by incorporating signal attenuation characteristics into the relaxation constraints, thereby preserving the nonlinear relationship between RSS and distance even under severe shadowing. In Step 2, a dynamic weighting mechanism is developed to refine the estimate. Unlike prior static weighting schemes, our weights adaptively combine Euclidean distance (to emphasize nearby anchors) and noise variance (to suppress high-noise measurements), enabling real-time adjustment based on the initial coarse estimate from Step 1. This dual-weighting strategy, coupled with the GTRS framework, effectively mitigates both relaxation-induced errors and measurement noise. The interplay between these steps ensures that the proposed method maintains robustness against signal attenuation while recovering the optimality lost in convex approximations.

The primary contributions of this paper are as follows:

(1) A nonconvex estimator is derived that approximates the ML estimator while eliminating the logarithmic term in the residual. A convex estimator is then obtained through convex relaxation.

(2) A weight is calculated based on the Euclidean distance and standard deviation, using the initial estimate from Step 1.

(3) The accuracy of the estimate is further refined using WLS and GTRS methods to recover performance degradation from the relaxation and approximation processes.

Throughout this paper, the following notations are used: $R^{n}$ denotes the set of *n*-dimensional real column vectors. The boldface lowercase letter $x\in R^{n}$ represents a column vector, while the boldface uppercase letter $A\in R^{M\times N}$ represents an M×N matrix. The entry $d_{i}$ denotes the *i*-th element of the vector d. The notation ∥·∥ refers to the $\ell_{2}$-norm, and $\max_{1⩽i⩽N}\left|b_{i}\right|$ represents the $\ell_{\infty }$-norm, also known as the Chebyshev norm. The transpose of vector a is denoted as $a^{T}$, and V(a) represents the variance of variable *a*. The operator diag(M) denotes the diagonal matrix with M as its diagonal elements, and $I_{N}$ represents an N×N identity matrix. For Hermitian matrices A and B, the notation A⪰B indicates that A−B is positive semidefinite.

The remainder of this paper is organized as follows. Section 2 introduces the RSS models, the localization scenario, and the localization problem. Section 3 outlines the derivation of the proposed two-step localization method. Section 4 discusses the computational complexity of the proposed method. Section 5 presents computer simulation results and compares the performance of the proposed method with other approaches. Finally, Section 6 concludes the paper.

## 2. System Model and Problem Formulation

Consider a WSN consisting of *N* anchor nodes and a target node. The anchor nodes are randomly deployed at known locations $s_{1},s_{2},\ldots ,s_{N}$ ($s_{i}={\left(s_{i1},s_{i2}\right)}^{T}\in R^{2}$), while the target node’s location $x={\left(x_{1},x_{2}\right)}^{T}\in R^{2}$ is unknown. A designated reference node is positioned at a known reference distance $d_{0}$ from the target node. It provides the reference path loss $L_{0}$ (in dB) at $d_{0}$, which serves as a baseline for signal strength normalization. Using this reference, the *i*-th anchor node can determine the path loss $L_{i}$ between itself and the target node, as illustrated in Figure 1. The corresponding measurement model is expressed as follows:(1)$$\begin{array}{c} L_{i}=L_{0}+10\gamma \log_{10}\left(\frac{∥x-s_{i}∥}{d_{0}}\right)+n_{i}, \end{array}$$
where γ is the path loss exponent (PLE), and $n_{i}$ represents log-normal shadowing terms (herein termed noise), which are modeled as $n_{i}∼N\left(0,\sigma_{n_{i}}^{2}\right)$.

Given the RSS measurements $L_{i}$ (i=1,2,…,N), the goal of RSS-based localization is to estimate the target’s location x using the model (Equation 1). The ML estimator of the target’s location is derived by minimizing the following expression:(2)$$\begin{array}{c} \min_{x}\sum_{i=1}^{N}{\left(L_{i}-L_{0}-10\gamma \log_{10}\left(\frac{∥x-s_{i}∥}{d_{0}}\right)\right)}^{2}. \end{array}$$

The goal is to obtain the most accurate estimate of the target node’s location, x. However, it is clear that (Equation 2) is nonconvex, posing significant challenges in its solution.

## 3. The Proposed Localization Method

In this section, we will detail the implementation of the proposed two-step SD/SOCP-GTRS method. The methodology is structured into two steps:

Step 1 (Convex Relaxation): The nonconvex ML problem is transformed into a convex problem by applying SDP and SOCP relaxation. This step generates an initial coarse estimate of the target location.

Step 2 (Refinement): Measurements are dynamically weighted based on their Euclidean distance and noise variance. The weighted data are then processed using the WLS-GTRS framework to refine the initial estimate, converging toward the global optimum.

The framework of the proposed scheme is illustrated in Figure 2.

### 3.1. Step 1: Convex Relaxation

To derive a convex estimator, the following lemmas are introduced.

**Lemma** **1.**
*If $x\in R^{d}$, then*

*(1) ${∥x∥}_{\infty }⩽{∥x∥}_{2}⩽\sqrt{d}{∥x∥}_{\infty }$;*

*(2) ${∥x∥}_{\infty }⩽{∥x∥}_{1}⩽d{∥x∥}_{\infty }$.*


**Lemma** **2.**
*The norms ${∥x∥}_{a}$ and ${∥x∥}_{b}$ are equivalent if there exist positive constants $c_{1}$ and $c_{2}$, such that $c_{1}{∥x∥}_{b}⩽{∥x∥}_{a}⩽c_{2}{∥x∥}_{b}$ for any $x\in R^{d}$.*


**Lemma** **3.**
*For any m,n∈R, $\max\left\{m,n\right\}⩾\frac{m+n}{2}$.*


Next, the proposed localization method is derived.

First, (Equation 1) is rewritten as follows:(3)$$\begin{array}{c} \frac{L_{i}-L_{0}}{5\gamma }=\log_{10}\left(\frac{∥x-s_{i}{∥}^{2}}{d_{0}^{2}}\right)+\frac{n_{i}}{5\gamma }. \end{array}$$

Using the change in base formula, (Equation 3) is expressed as follows:(4)$$\begin{array}{c} \frac{\ln10(L_{i}-L_{0})}{5\gamma }=\ln\left(\frac{∥x-s_{i}{∥}^{2}}{d_{0}^{2}}\right)+\frac{\ln10}{5\gamma }n_{i}. \end{array}$$

Applying the logarithmic identity, (Equation 4) becomes the following:(5)$$\begin{array}{c} \ln\left(e^{\frac{\ln10(L_{i}-L_{0})}{5\gamma }}\right)=\ln\left(\frac{∥x-s_{i}{∥}^{2}}{d_{0}^{2}}\right)+\frac{\ln10}{5\gamma }n_{i}. \end{array}$$

Let $\beta_{i}^{2}=d_{0}^{2}e^{\frac{\ln10(L_{i}-L_{0})}{5\gamma }}$, where e is the base of the natural logarithm. Ref. (Equation 5) is then transformed into the following:(6)$$\begin{array}{c} \ln\frac{\beta_{i}^{2}}{∥x-s_{i}{∥}^{2}}=\frac{\ln10}{5\gamma }n_{i}. \end{array}$$

Thus, the ML estimator is formulated as follows:(7)$$\begin{array}{c} \min_{x}\sum_{i=1}^{N}{\left(\ln\frac{\beta_{i}^{2}}{∥x-s_{i}{∥}^{2}}\right)}^{2}. \end{array}$$

By applying Lemmas 1 and 2 and replacing the $\ell_{2}$ norm with the $\ell_{\infty }$ norm (also known as the Chebyshev norm), the optimization problem (Equation 7) can be reformulated into a Chebyshev norm form:(8)$$\begin{array}{c} \min_{x}\;\max_{1⩽i⩽N}|\ln\frac{\beta_{i}^{2}}{∥x-s_{i}{∥}^{2}}|. \end{array}$$

Then,(9)$$\begin{array}{c} |\ln\frac{\beta_{i}^{2}}{∥x-s_{i}{∥}^{2}}|=\left\{\begin{array}{c} \ln\frac{\beta_{i}^{2}}{∥x-s_{i}{∥}^{2}},\;\frac{\beta_{i}^{2}}{∥x-s_{i}{∥}^{2}}⩾1,\\ \ln\frac{∥x-s_{i}{∥}^{2}}{\beta_{i}^{2}},\;0<\frac{\beta_{i}^{2}}{∥x-s_{i}{∥}^{2}}<1. \end{array}\right. \end{array}$$

The following can then be derived:(10)$$\begin{array}{c} \max_{1⩽i⩽N}|\ln\frac{\beta_{i}^{2}}{∥x-s_{i}{∥}^{2}}|=\max_{1⩽i⩽N}\ln\left(\max\left\{\frac{\beta_{i}^{2}}{∥x-s_{i}{∥}^{2}},\frac{∥x-s_{i}{∥}^{2}}{\beta_{i}^{2}}\right\}\right). \end{array}$$

Substituting (Equation 10) into (Equation 8) results in the following:(11)$$\begin{array}{c} \min_{x}\;\max_{1⩽i⩽N}\ln\left(\max\left\{\frac{\beta_{i}^{2}}{∥x-s_{i}{∥}^{2}},\frac{∥x-s_{i}{∥}^{2}}{\beta_{i}^{2}}\right\}\right). \end{array}$$

Since f(x)=lnx is a strictly monotonic function within its domain, (Equation 11) can be simplified by eliminating the logarithm, resulting in the following form:(12)$$\begin{array}{c} \min_{x}\;\max_{1⩽i⩽N}\left(\max\left\{\frac{\beta_{i}^{2}}{∥x-s_{i}{∥}^{2}},\frac{∥x-s_{i}{∥}^{2}}{\beta_{i}^{2}}\right\}\right). \end{array}$$

Using Lemma 3, the following inequality is obtained:(13)$$\begin{array}{c} \max\left\{\frac{\beta_{i}^{2}}{∥x-s_{i}{∥}^{2}},\frac{∥x-s_{i}{∥}^{2}}{\beta_{i}^{2}}\right\}⩾\frac{\frac{\beta_{i}^{2}}{∥x-s_{i}{∥}^{2}}+\frac{∥x-s_{i}{∥}^{2}}{\beta_{i}^{2}}}{2}. \end{array}$$

Thus, the (Equation 12) can be reformulated as follows:(14)$$\begin{array}{c} \min_{x}\;\max_{1⩽i⩽N}\left(\frac{\frac{\beta_{i}^{2}}{∥x-s_{i}{∥}^{2}}+\frac{∥x-s_{i}{∥}^{2}}{\beta_{i}^{2}}}{2}\right). \end{array}$$

By applying Lemmas 1 and 2 and substituting the $\ell_{\infty }$ norm with the $\ell_{2}$ norm, the optimization problem (Equation 14) is expressed as follows:(15)$$\begin{array}{c} \min_{x}\;\sum_{i=1}^{N}{\left(\frac{\frac{\beta_{i}^{2}}{∥x-s_{i}{∥}^{2}}+\frac{∥x-s_{i}{∥}^{2}}{\beta_{i}^{2}}}{2}\right)}^{2}. \end{array}$$

Introducing auxiliary variables $u_{i}=\frac{\beta_{i}^{2}}{∥x-s_{i}{∥}^{2}}$, $v_{i}=\frac{∥x-s_{i}{∥}^{2}}{\beta_{i}^{2}}$ and $z_{i}=\frac{u_{i}+v_{i}}{2}$, (Equation 15) can be further expressed as follows:(16a)$$\begin{array}{cc} \; & \min_{\begin{array}{c} x,z,u_{i},v_{i} \end{array}}\;{∥z∥}^{2}, \end{array}$$
s.t.
(16b)$$\begin{array}{cc} \; & z_{i}=\frac{u_{i}+v_{i}}{2}, \end{array}$$(16c)$$\begin{array}{cc} \; & u_{i}=\frac{\beta_{i}^{2}}{∥x-s_{i}{∥}^{2}}, \end{array}$$(16d)$$\begin{array}{cc} \; & v_{i}=\frac{∥x-s_{i}{∥}^{2}}{\beta_{i}^{2}}. \end{array}$$

To solve (Equation 14) more effectively, an epigraph variable *t* is introduced, and semidefinite and second-order cone relaxations are applied in the forms of $1⩾u_{i}v_{i}$ and ${∥z∥}^{2}⩽t$, respectively. As a result, the optimization problem (16) is transformed into the following epigraph form:(17a)$$\begin{array}{cc} \; & \min_{\begin{array}{c} x,z,u_{i},v_{i} \end{array},t}\;t, \end{array}$$
s.t.(17b)$$\begin{array}{cc} \; & z_{i}=\frac{u_{i}+v_{i}}{2}, \end{array}$$(17c)$$\begin{array}{cc} \; & u_{i}=\frac{\beta_{i}^{2}}{∥x-s_{i}{∥}^{2}}, \end{array}$$(17d)$$\begin{array}{cc} \; & v_{i}=\frac{∥x-s_{i}{∥}^{2}}{\beta_{i}^{2}}, \end{array}$$(17e)$$\begin{array}{cc} \; & ∥\left[\begin{array}{c} 2z\\ t-1 \end{array}\right]∥⩽t+1, \end{array}$$(17f)$$\begin{array}{cc} \; & \left[\begin{array}{cc} u_{i} & 1\\ 1 & v_{i} \end{array}\right]⪰0. \end{array}$$

Next, let $y={∥x∥}^{2}$. Using the relationship between SDP and SOCP, we can express the nonconvex constraint $y={∥x∥}^{2}$ as ${∥x∥}^{2}⩽y$. This allows the problem (17) to be reformulated as the following convex problem:(18a)$$\begin{array}{cc} \; & \min_{\begin{array}{c} x,z,u_{i},v_{i} \end{array},t,y}\;t, \end{array}$$
s.t.(18b)$$\begin{array}{cc} \; & z_{i}=\frac{u_{i}+v_{i}}{2}, \end{array}$$(18c)$$\begin{array}{cc} \; & \left[\begin{array}{cc} u_{i} & \beta_{i}\\ \beta_{i} & y-2s_{i}^{T}x+{∥s_{i}∥}^{2} \end{array}\right]⪰0, \end{array}$$(18d)$$\begin{array}{cc} \; & v_{i}=\frac{y-2s_{i}^{T}x+{∥s_{i}∥}^{2}}{\beta_{i}^{2}}, \end{array}$$(18e)$$\begin{array}{cc} \; & ∥\left[\begin{array}{c} 2z\\ t-1 \end{array}\right]∥⩽t+1, \end{array}$$(18f)$$\begin{array}{cc} \; & \left[\begin{array}{cc} u_{i} & 1\\ 1 & v_{i} \end{array}\right]⪰0, \end{array}$$(18g)$$\begin{array}{cc} \; & ∥\left[\begin{array}{c} 2x\\ y-1 \end{array}\right]∥⩽y+1. \end{array}$$

Upon solving the convex problem (18), a preliminary estimate of the target’s location, denoted as $x_{\mathrm{cvx}}$, is obtained. This estimator is referred to as “SD/SOCP”.

### 3.2. Step 2: Refinement

In Step 1, the effects of noise and distance on localization performance are not considered. In this step, the approximate solution $x_{\mathrm{cvx}}$ obtained in Step 1 is utilized to account for the impact of log-normal shadowing terms. Additionally, weights are introduced to emphasize nearby links, incorporating the influence of the distance between anchor nodes and the target node to further enhance localization accuracy.

First, note that (Equation 6) can be reformulated as follows:(19)$$\begin{array}{c} \ln\frac{∥x-s_{i}{∥}^{2}}{\beta_{i}^{2}}=-\frac{\ln10}{5\gamma }n_{i}. \end{array}$$

Next, a first-order Taylor series approximation is applied to the logarithmic terms, leading to the following expression:(20)$$\begin{array}{c} ∥x-s_{i}{∥}^{2}-\beta_{i}^{2}\approx \xi_{i}, \end{array}$$
where $\xi_{i}=-\frac{\beta_{i}^{2}\ln10}{5\gamma }n_{i}$.

The Variance of $\xi_{i}$ is calculated as follows:(21)$$\begin{array}{c} V\left(\xi_{i}\right)=E\left({\left(\xi_{i}-E\left(\xi_{i}\right)\right)}^{2}\right)={\left(\frac{\beta_{i}^{2}\ln10}{5\gamma }\right)}^{2}\sigma_{i}^{2}. \end{array}$$

For simplicity, assume that $\sigma_{i}=\sigma$ for all anchor nodes. In this case, the weight $w_{1i}$ can be expressed as follows:(22)$$\begin{array}{c} w_{1i}=\frac{\sigma^{2}}{V(\xi_{i})}=\frac{1}{{\left(\frac{\beta_{i}^{2}\ln10}{5\gamma }\right)}^{2}}. \end{array}$$

To further enhance the localization accuracy, an additional weight $w_{2i}$ is introduced to prioritize nearby links:(23)$$\begin{array}{c} w_{2i}=1-\frac{{\hat{d}}_{i}}{\sum_{i=1}^{N}{\hat{d}}_{i}}, \end{array}$$
where ${\hat{d}}_{i}=∥x_{\mathrm{cvx}}-s_{i}∥$, and $x_{\mathrm{cvx}}$ is the target node’s estimated location from Step 1. This weighting scheme is based on the fact that RSS short-range measurements are more reliable than long-range ones. The RSS measurements have a constant multiplicative factor with distance, leading to larger errors for remote links compared to nearby ones. Thus, this weight definition better reflects the reliability of different links, improving localization accuracy.

The final comprehensive weight is then calculated as follows:(24)$$\begin{array}{c} w_{i}=\sqrt{w_{1i}w_{2i}}. \end{array}$$

Thus, the WLS estimator becomes the following:(25)$$\begin{array}{c} \min_{x}\;\sum_{i=1}^{N}w_{i}{\left(∥x-s_{i}{∥}^{2}-\beta_{i}^{2}\right)}^{2}. \end{array}$$

By introducing the substitution $\tilde{x}={\left[x,∥x∥\right]}^{T}$, the optimization problem (Equation 27) can be reformulated into a GTRS form:(26a)$$\begin{array}{cc} \min_{\tilde{x}}\; & ∥W\left(A\tilde{x}-b\right){∥}^{2}, \end{array}$$
s.t.(26b)$$\begin{array}{cc} \; & {\tilde{x}}^{T}D\tilde{x}+2\iota^{T}\tilde{x}=0, \end{array}$$
where D=diag([1,1,0,0]), $\iota ={\left[0,0,-\frac{1}{2}\right]}^{T}$, $W=\mathrm{diag}(\left[w_{i}\right])$, and $A=\left[\begin{array}{cc} -2s_{1}^{T} & 1\\ \vdots & \vdots \\ -2s_{N}^{T} & 1 \end{array}\right]$, $b=\left[\begin{array}{c} \beta_{1}^{2}-{∥s_{1}∥}^{2}\\ \vdots \\ \beta_{N}^{2}-{∥s_{N}∥}^{2} \end{array}\right]$.

The refinement step involves using the estimate from Step 1 to calculate the weights, giving more importance to nearby links and accounting for the noise variance at each anchor node to improve the accuracy of the target’s location estimation. Simulation results in Section 5 demonstrate the effectiveness of the proposed method. In the following sections, we refer to the convex problem (26) as “SD/SOCP-GTRS”.

Figure 3 illustrates the two-step localization process. For (a), the original nonconvex ML problem (red curve) exhibits multiple local minima, making it challenging to find the global optimum. This complexity is due to the nonlinear effects of signal attenuation and noise. For (b), convex relaxation (blue region) transforms the nonconvex problem into a convex problem by expanding the feasible space. This step simplifies the search space, enabling an efficient global search for an initial estimate ($x_{\mathrm{cvx}}$). Although this may introduce slight bias, it ensures a feasible initial estimate even under severe noise conditions. For (c), refinement via WLS-GTRS (green arrow) adjusts the initial estimate toward the true ML optimum. This refinement prioritizes reliable measurements by dynamically weighting based on Euclidean distance and noise variance, effectively ’pulling’ the solution closer to the true target location. The interplay between these steps ensures robustness against signal attenuation and noise while recovering the optimality lost in convex approximations.

## 4. Computational Complexity Analysis

This section analyzes the computational complexity of the proposed SD/SOCP-GTRS method and compares it with existing approaches. Grasping the computational demands of each step is vital for evaluating the method’s practicality and efficiency. All considered methods involve an inherent trade-off between estimation accuracy and implementation complexity. Here, the method for the worst-case complexity of the mixed SD/SOCP, as presented in [37], is adopted to analyze the complexities of the proposed method and other methods considered in this paper. The computational complexity formula is as follows:(27)$$\begin{array}{c} O\left(\sqrt{L}\left(m\sum_{i=1}^{N_{sdc}}n_{i}^{sdc^{3}}+m^{2}\sum_{i=1}^{N_{sdc}}n_{i}^{sdc^{2}}+m^{2}\sum_{i=1}^{N_{soc}}n_{i}^{soc}+\sum_{i=1}^{N_{soc}}n_{i}^{soc^{2}}+m^{3}\right)\right), \end{array}$$
where *L* represents the number of iterations of the method, *m* is the number of equality constraints, $N_{sdc}$ and $N_{soc}$ are the numbers of semi-definite cone (SDC) and second-order constraints (SOC), respectively, and $n_{i}^{sdc}$ and $n_{i}^{soc}$ are the dimensions of the *i*-th SDC and *i*-th SOC, respectively. Assume $K_{\max}$ is the maximum number of steps in the bisection procedure used by GTRS.

For the proposed SD/SOCP-GTRS method, in Step 1, it needs to solve a SOCP with two constraints and an SDP with 2N constraints. The worst-case complexity, mainly determined by the interior point algorithm, scales as $O(N^{3.5})$. This step is crucial as it transforms the nonconvex problem into a tractable convex form, thus providing a reliable initial estimate. In Step 2, GTRS has to solve a quadratically constrained quadratic program. With the bisection method involving $K_{\max}$ iterations, the complexity is $O(K_{\max}N)$. This refinement step improves the accuracy of the initial estimate by adaptively weighing measurements according to their reliability. Consequently, the overall complexity of the proposed method is the sum of the complexities of these two steps, namely $O\left(N^{3.5}\right)+O\left(K_{\max}N\right)$.

Table 1 summarizes the worst-case complexities of each method. The proposed SD/SOCP-GTRS method strikes a favorable balance between computational complexity and localization accuracy. By effectively integrating convex relaxation with a refinement step, it attains high accuracy while maintaining manageable computational requirements. This makes it especially suitable for real-time applications in WSNs and other resource-constrained scenarios.

## 5. Simulation Results

In this section, we evaluate the performance of the proposed SD/SOCP-GTRS estimator through Monte Carlo (Mc) simulations. We simulate a relatively small environment, such as a home or a small office, where one target node is randomly and uniformly distributed within a square area of 20×20 $m^{2}$. Figure 4 shows an example of a network deployed with one target node and five anchor nodes. We examine networks with two different anchor node deployments. In the first network, the anchor nodes are regularly placed at the corners and the center of the area, as shown in Figure 4a. In the second network, the anchor nodes are irregularly placed, as depicted in Figure 4b. In the simulations, unless otherwise specified, in the simulated experiments, the positions of the target node and the anchor nodes are irregularly and randomly distributed within the square area of 20×20 $m^{2}$. Initially, it is assumed that the network is fully connected; that is, all anchor nodes and the target node can communicate with each other. The path loss exponent of the RSS measurement model (Equation 1) is set to γ=2.5, and the shadowing standard deviation is set to σ=4 dB. It is assumed that all nodes have the same transmission power, and the received power at the reference distance $d_{0}=1$ m is set to $L_{0}=40$ dB. We use the CVX toolbox to solve the SDP and SOCP problems [34,35,37,38,39,40] and use MATLAB’s *lsqnonlin* function to solve the ML estimator. The proposed method is compared with three other algorithms in the literature, and these algorithms are described in Table 1. For context, we also include the CRLB and the ML solution using the true target location as the starting point, referred to as “ML-True”. In the following sections, we will assess the performance of the proposed method across various scenarios. The performance of the discussed methods is evaluated using the root mean square error (RMSE), defined as follows:(28)$$\begin{array}{c} \mathrm{RMSE}=\sqrt{\frac{1}{\mathrm{Mc}}\sum_{i=1}^{\mathrm{Mc}}{∥x-\hat{x}∥}^{2}}, \end{array}$$
where $\hat{x}$ and x represent the estimated and true locations for the *i*-th Monte Carlo run, respectively, and Mc=3000 denotes the number of runs.

### 5.1. Effect of the Shadowing Standard Deviation

Figure 5 shows the RMSE performance of different localization methods under different shadowing standard deviations (σ). As σ increases from low to high values, the RMSE of all methods increases due to reduced RSS measurement reliability from higher noise levels. Specifically, the RMSE of the DEOR method shows a significant increase at higher σ values, while the proposed SD/SOCP-GTRS method achieves a notable reduction in error compared to others. Although the intermediate estimate of the first step (SD/SOCP) remains stable with a certain level of RMSE, it is slightly less accurate than the SDP-LSRE method. However, after the second step (GTRS refinement), the RMSE of SD/SOCP-GTRS further decreases, outperforming others and staying close to the theoretical bounds across all σ values. This indicates that the proposed method can maintain robustness through two-step optimization even under strong noise interference, with performance approaching the theoretical optimum.

### 5.2. Effect of the Number of the Anchor Nodes

Figure 6 shows how the number of anchor nodes (*N*) affects localization accuracy when σ=4 dB and γ=2.5. As *N* increases from 4 to 16, the RMSE of all methods decreases significantly. For instance, the RMSE of our SD/SOCP-GTRS method drops substantially with increasing *N*, showing a marked improvement in localization accuracy. More anchor nodes provide redundant RSS measurements, and their spatial diversity effectively suppresses noise. Notably, SD/SOCP-GTRS outperforms other methods (except ML-True) across all anchor node configurations. At moderate *N* values, its RMSE is significantly lower than SDP’s and remains close to the theoretical bounds. This indicates that SD/SOCP-GTRS remains robust, whether in sparse or dense anchor node deployments, offering a flexible precision-cost trade-off for real-world networks like urban environments or industrial facilities.

## 6. Conclusions

This paper presents a two-step localization method for RSS-based localization in WSNs. In the first step, a convex estimator is derived using SDP and SOCP. In the second step, a weight is introduced to further enhance localization accuracy. Monte Carlo simulations are conducted to evaluate the effects of shadowing standard deviation and the number of anchor nodes. The results demonstrate that the proposed SD/SOCP-GTRS method consistently outperforms the other methods, establishing it as the optimal choice for RSS-based localization in WSNs. This method has practical deployment potential in critical applications such as smart cities, enabling precise tracking of vehicles and pedestrians. Future work will extend it to 3D environments and integrate hybrid measurement systems to enhance the robustness of 6G-enabled smart infrastructures. Additionally, the impact of non-uniform anchor node distribution on localization accuracy will be studied, and the integration of this method with other localization techniques, such as AOA or TDOA, will be explored.

## Figures and Tables

**Figure 1 sensors-25-01837-f001:**
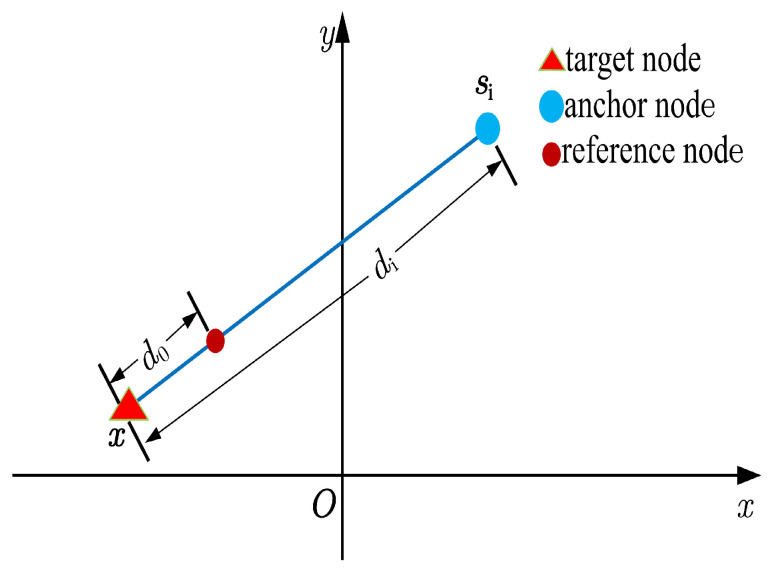
Illustration of the link between the target node and the *i*-th anchor node in WSNs.

**Figure 2 sensors-25-01837-f002:**
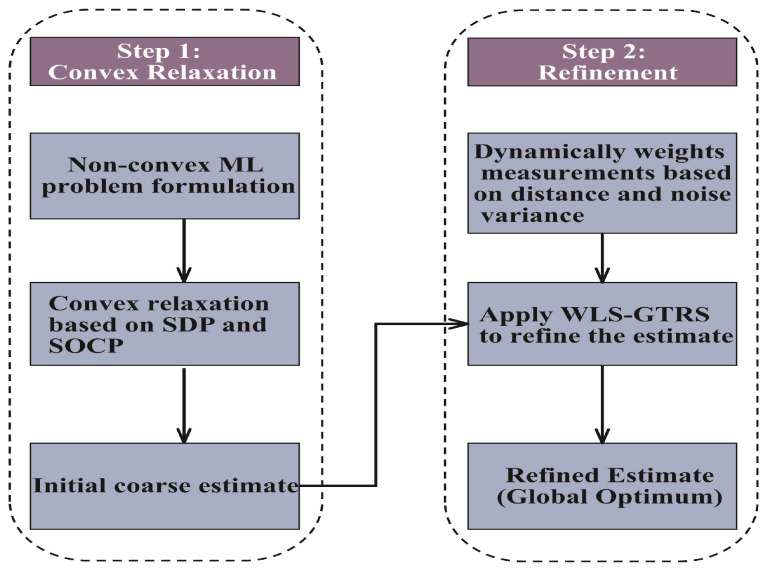
Proposed two-step SD/SOCP-GTRS method flowchart.

**Figure 3 sensors-25-01837-f003:**
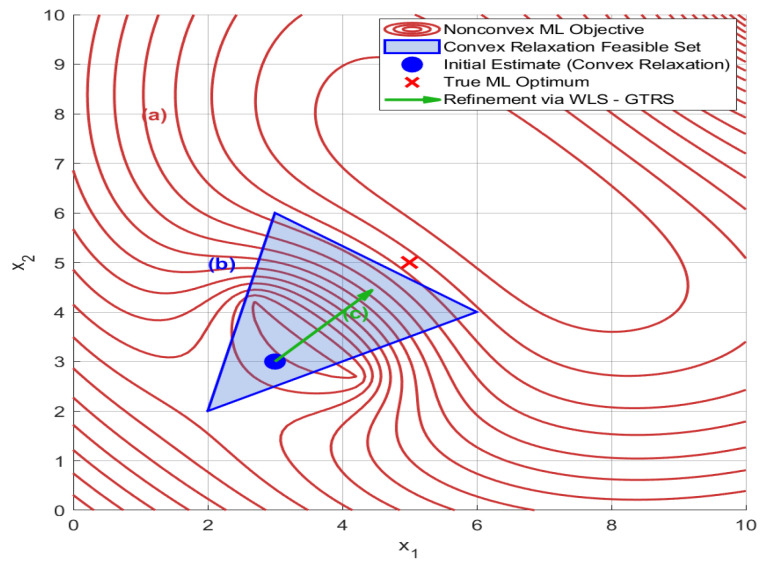
Geometric interpretation of convex relaxation and refinement for the nonconvex ML problem.

**Figure 4 sensors-25-01837-f004:**
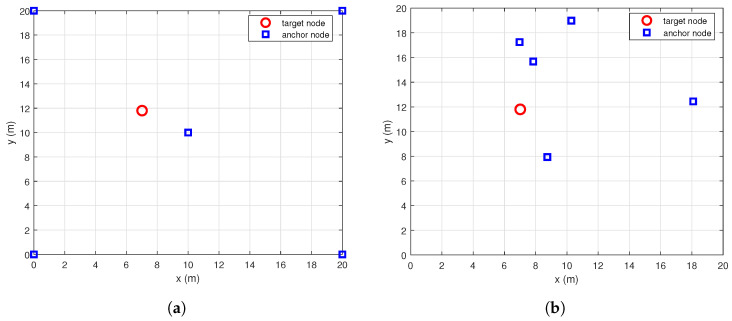
Two networks with different anchor nodes deployments. The target node and anchor nodes are represented by circles and squares, respectively. (**a**) The first network with a regular deployment. (**b**) The second network with an irregular deployment.

**Figure 5 sensors-25-01837-f005:**
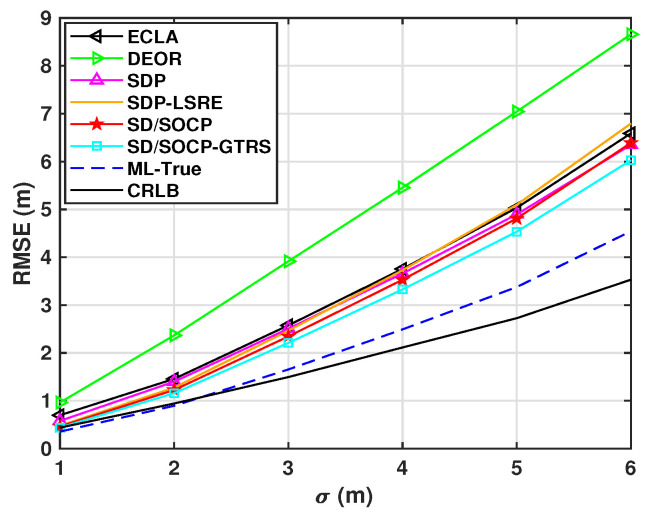
RMSE versus shadowing standard deviation σ (dB) for N=8, γ=2.5, $L_{0}=40$ dB, $d_{0}=1$ m, and Mc=3000.

**Figure 6 sensors-25-01837-f006:**
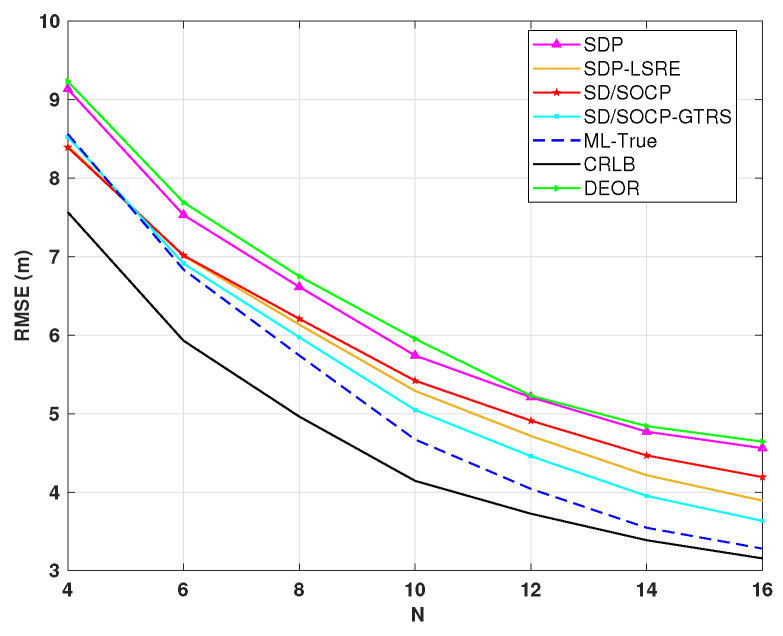
RMSE versus the number of anchor nodes *N* for σ=4 dB, γ=2.5, $L_{0}=40$ dB, $d_{0}=1$ m, and Mc=3000.

**Table 1 sensors-25-01837-t001:** Worst-case computational complexities of considered methods.

Method	Description	Complexity
SDP	The SDP method in [19]	$O(N^{4.5})$
SDP-LSRE	The SDP-LSRE method in [20]	$O(N^{4.5})$
ECLA	The ECLA method in [36]	$O(K_{max}N)$
DEOR	The DEOR method in [33]	$O(K_{max}N)$
SD/SOCP	The proposed method in Section 3.1	$O(N^{3.5})$
SD/SOCP-GTRS	The proposed method in Section 3.2	$O\left(N^{3.5}\right)+O\left(K_{max}N\right)$

## Data Availability

No new data were created or analyzed in this study. Data sharing is not applicable to this article.

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
