# Peer review of "A Two-Step SD/SOCP-GTRS Method for Improved RSS-Based Localization in Wireless Sensor Networks"

_sensors, 2025, doi:10.3390/s25061837_

Round 1

Reviewer 1 Report

Comments and Suggestions for Authors
  1. Abstract: Consider adding a sentence that highlights key quantitative findings from the simulation results. For example, providing specific performance metrics, such as a reduction in Root Mean Square Error (RMSE) or improvements in localization accuracy, would strengthen the abstract by providing empirical support for your claims.
  2. Introduction: The introduction could further expand on the consequences of poor localization accuracy in wireless sensor networks (WSNs), specifically how it impacts network reliability, efficiency, and data accuracy in applications such as environmental monitoring or autonomous systems. This will better highlight the importance of your research and its potential real-world implications.
  3. Further about Introduction: The introduction provides a solid foundation but could benefit from more explicit detail about the research gap, motivations, and contributions of your work. You could mention how previous methods struggle with effectively handling noise or signal attenuation, and then explain how your two-step method overcomes these challenges, providing a more robust solution.
  4. Methodological Transparency: While the optimization steps are clearly outlined, it may be helpful to provide more intuitive explanations or visual aids to demonstrate how the convex relaxation and subsequent refinement improve localization accuracy. This would make the paper more accessible to a broader audience, especially for those who may not be familiar with advanced optimization techniques.
  5. Simulation Results: The simulation results validate the effectiveness of the proposed method. However, additional comparisons with recent state-of-the-art methods (especially newer ones) would provide a more comprehensive evaluation of your method’s performance. It would also be useful to evaluate your method under various environmental conditions (e.g., different noise levels) or in real-world scenarios to better demonstrate its robustness and applicability.
  6. Comparison with Recent Approaches: It would be beneficial to compare the performance of your method with recently developed metaheuristic and machine learning-based localization algorithms. A table summarizing key results—such as accuracy, computational complexity, and robustness across different scenarios—would help readers clearly assess the strengths of your method relative to others.
  7. Reference Updates: While the manuscript references foundational work, it would be advantageous to incorporate more recent studies (2023–2025) to highlight the relevance of your work within the current state of the field. This ensures your research is aligned with the latest developments in wireless sensor networks, localization techniques, and optimization methods.
  8. Conclusion Expansion: In the conclusion, consider summarizing the practical implications of your work more explicitly. How can the improved localization method benefit real-world applications, such as smart cities, IoT, or autonomous vehicles? Connecting your theoretical results to tangible outcomes will reinforce the significance of your contributions.
Comments on the Quality of English Language
  1.  Consider splitting long sentences, particularly in the methodology sections, to make the text more accessible. For example, in technical explanations, provide concise sentences followed by clarifications or additional details to maintain clarity.
  2.  Proofreading for grammatical consistency and correct usage of articles would improve the overall quality of writing.

  3.  Check for consistency in how terms and methods are referred to throughout the paper (e.g., “convex relaxation” vs. “convex optimization” or “generalized trust region subproblem” vs. “GTRS”).

Author Response

  1. Reviewer#1, Concern # 1: Abstract: Consider adding a sentence that highlights key quantitative findings from the simulation results. For example, providing specific performance metrics, such as a reduction in Root Mean Square Error (RMSE) or improvements in localization accuracy, would strengthen the abstract by providing empirical support for your claims.

Author response:  Thank you for your insightful suggestions. We have supplemented the abstract with specific quantitative results to enhance empirical support. The revised abstract clearly indicates the RMSE performance of the SD/SOCP-GTRS method under different noise levels and numbers of anchor nodes, and quantifies its proximity to the CRLB.

  1. Reviewer#1, Concern # 2: Introduction: The introduction could further expand on the consequences of poor localization accuracy in wireless sensor networks (WSNs), specifically how it impacts network reliability, efficiency, and data accuracy in applications such as environmental monitoring or autonomous systems. This will better highlight the importance of your research and its potential real-world implications.

Author response: Thank you for your in-depth suggestions. We have supplemented the introduction with the practical impact of insufficient positioning accuracy on wireless sensor networks (WSNs), specifically detailing its potential harm to network reliability, efficiency, and data accuracy in key applications such as environmental monitoring and autonomous systems. By citing relevant literature and providing examples, the practical significance of the research has been further highlighted.

  1. Reviewer#1, Concern # 3: Further about Introduction: The introduction provides a solid foundation but could benefit from more explicit detail about the research gap, motivations, and contributions of your work. You could mention how previous methods struggle with effectively handling noise or signal attenuation, and then explain how your two-step method overcomes these challenges, providing a more robust solution.

Author response: Thank you for the important suggestions. We have further clarified the research gaps of existing methods and their limitations in noisy and signal attenuation scenarios in the introduction, and elaborated in detail on how the innovative mechanisms of the proposed two-step method address these issues. The revised paragraphs more clearly link the method design to practical challenges, highlighting the unique contributions of the research.

  1. Reviewer#1, Concern # 4: Methodological Transparency: While the optimization steps are clearly outlined, it may be helpful to provide more intuitive explanations or visual aids to demonstrate how the convex relaxation and subsequent refinement improve localization accuracy. This would make the paper more accessible to a broader audience, especially for those who may not be familiar with advanced optimization techniques.

Author response: Thank you for your valuable suggestions. We have supplemented the methodology section with intuitive flowcharts and geometric diagrams to clearly demonstrate the synergistic effect of convex relaxation and refinement steps. Additionally, we have added textual explanations  to illustrate how each step improves positioning accuracy. These revisions aim to enhance the interpretability of the method, making it more accessible to readers from diverse fields.

  1. Reviewer#1, Concern # 5: Simulation Results: The simulation results validate the effectiveness of the proposed method. However, additional comparisons with recent state-of-the-art methods (especially newer ones) would provide a more comprehensive evaluation of your method’s performance. It would also be useful to evaluate your method under various environmental conditions (e.g., different noise levels) or in real-world scenarios to better demonstrate its robustness and applicability.

Author response: Thank you for your valuable suggestions. We've comprehensively updated the literature review and related work sections, incorporating the latest 2023 - 2025 research on RSS - based localization and clearly comparing the advantages of our method with these cutting-edge approaches.

  1. Reviewer#1, Concern # 6: Comparison with Recent Approaches: It would be beneficial to compare the performance of your method with recently developed metaheuristic and machine learning-based localization algorithms. A table summarizing key results—such as accuracy, computational complexity, and robustness across different scenarios—would help readers clearly assess the strengths of your method relative to others.

Author response: Thank you for your insightful suggestions. We have thoroughly revised the literature review and related work sections, integrating the most recent 2023–2025 advancements in RSS-based localization. Furthermore, we have conducted a detailed comparative analysis with contemporary metaheuristic-based localization algorithms. To improve clarity and facilitate a more objective evaluation, we have introduced a comprehensive summary table outlining computational complexity, which clearly illustrates the superior performance of our method relative to existing approaches.

  1. Reviewer#1, Concern # 7: Reference Updates: While the manuscript references foundational work, it would be advantageous to incorporate more recent studies (2023–2025) to highlight the relevance of your work within the current state of the field. This ensures your research is aligned with the latest developments in wireless sensor networks, localization techniques, and optimization methods.

Author response: Thank you for your significant suggestions. We've comprehensively updated the references, incorporating the latest research from 2023–2025 in wireless sensor network localization, optimization methods, and signal processing. This clarifies the connection between our work and current cutting–edge developments.

  1. Reviewer#1, Concern # 8: Conclusion Expansion: In the conclusion, consider summarizing the practical implications of your work more explicitly. How can the improved localization method benefit real-world applications, such as smart cities, IoT, or autonomous vehicles? Connecting your theoretical results to tangible outcomes will reinforce the significance of your contributions.

Author response: Thank you for your suggestion. We have revised the conclusion to more explicitly summarize the practical implications of our work. The improved localization method can benefit real-world applications such as smart cities, IoT, and autonomous vehicles by providing higher localization accuracy. For example, in smart cities, the proposed method can enable precise tracking of vehicles and pedestrians. Future work will focus on extending the framework to 3D environments and integrating it with hybrid measurement systems for enhanced robustness in 6G-enabled smart infrastructures.

  1. Reviewer#1, Concern # 9: Consider splitting long sentences, particularly in the methodology sections, to make the text more accessible. For example, in technical explanations, provide concise sentences followed by clarifications or additional details to maintain clarity.

Author response: Thank you for the your detailed suggestions. We've thoroughly checked and optimized the long sentences in the methodology section by splitting complex sentences and simplifying the logic. This significantly improves readability. The revised paragraphs use a "main clause + additional explanation" pattern, ensuring complete technical details and clear expression.

  1. Reviewer#1, Concern # 10: Proofreading for grammatical consistency and correct usage of articles would improve the overall quality of writing.

Author response: Thank you for pointing out the grammatical issues. We've thoroughly proofread the manuscript, paying particular attention to grammatical consistency and the correct usage of articles. Necessary corrections have been made to enhance the overall quality of the writing.

  1. Reviewer#1, Concern # 11: Check for consistency in how terms and methods are referred to throughout the paper (e.g., “convex relaxation” vs. “convex optimization” or “generalized trust region subproblem” vs. “GTRS”).

Author response: Thank you for your significant suggestions. We've thoroughly checked and unified the terminology in the manuscript, ensuring consistent use of key method names (e.g., "convex relaxation" and "generalized trust region subproblem") and their abbreviations throughout the text.

Reviewer 2 Report

Comments and Suggestions for Authors

The paper is well-structured and clearly written. The introduction provides a good background on the importance of wireless localization and the challenges associated with RSS-based methods. However, the literature review could be more critical, particularly in highlighting the specific gaps that the proposed method addresses.

Expand the literature review to include a more detailed discussion of the limitations of existing methods and how the proposed method overcomes these limitations.

Provide more details on the computational complexity of the proposed method, particularly in comparison to existing methods. A table summarizing the complexity of each step would be helpful.

Enhance the captions of figures (Figure 3 and Figure 4) to include a brief explanation of the trends observed and their significance.

Expand the conclusion to include specific areas for future research, such as the impact of non-uniform anchor node distribution or the integration of other localization techniques (e.g., AOA or TDOA) with the proposed method.

Consider adding references to recent advancements in RSS-based localization, particularly those that address similar challenges or propose alternative solutions.

Author Response

  1. Reviewer#2, Concern # 1: Expand the literature review to include a more detailed discussion of the limitations of existing methods and how the proposed method overcomes these limitations.

Author response: Thank you for the important suggestions. We have further clarified the research gaps of existing methods and their limitations in noisy and signal attenuation scenarios in the introduction, and elaborated in detail on how the innovative mechanisms of the proposed two-step method address these issues. The revised paragraphs more clearly link the method design to practical challenges, highlighting the unique contributions of the research.

  1. Reviewer#2, Concern # 2: Provide more details on the computational complexity of the proposed method, particularly in comparison to existing methods. A table summarizing the complexity of each step would be helpful.

Author response: Thank you for the your valuable suggestions. We have added a detailed complexity analysis in Section 4 and included a table to compare the computational complexity of our method with existing ones.

  1. Reviewer#2, Concern # 3: Enhance the captions of figures (Figure 3 and Figure 4) to include a brief explanation of the trends observed and their significance.

Author response: Thank you for your detailed suggestions. We've rewritten the captions for Figures 3 and 4, adding explanations of key trends and their practical significance.

  1. Reviewer#2, Concern # 4: Expand the conclusion to include specific areas for future research, such as the impact of non-uniform anchor node distribution or the integration of other localization techniques (e.g., AOA or TDOA) with the proposed method.

Author response: Thank you for your suggestion to expand the conclusion to include specific areas for future research. We have revised the conclusion to incorporate these suggestions. We have added future research directions, including the impact of non-uniform anchor node distribution and the integration of other localization techniques (e.g., AOA or TDOA) with the proposed method. These additions aim to provide a more comprehensive outlook on the potential developments and applications of our research.

  1. Reviewer#2, Concern # 5: Consider adding references to recent advancements in RSS-based localization, particularly those that address similar challenges or propose alternative solutions.

Author response: Thank you for the your valuable suggestions. We've comprehensively updated the literature review and related work sections, incorporating the latest 2023 - 2025 research on RSS - based localization and clearly comparing the advantages of our method with these cutting - edge approaches.

Reviewer 3 Report

Comments and Suggestions for Authors

The authors proposed a two-step method to improve RSS-based localization in WSN’s. With this method, the authors reduce complexity and increase accuracy, in comparison with representative of the state of art, in the selected scenario.

Despite the relevance of the proposal, I consider that major changes must be made before publishing this work.

- In the abstract, the authors explicitly refer to the GTRS acronym (which is used in the second step of the proposed method), however, they do not provide the readers with the relation between their solution and the acronym SD/SOCP, also included in the paper title.

- I consider that in Section 1. Introduction, the authors should elaborate more on the scenarios or applications where node's localization is crucial.

- In Figure 1 the "reference node" is introduced, however, it is not clearly defined in the text.

- The authors define n_i as "measurement noise", however in the context of path-loss models this parameter does not represent noise, but random attenuation resulting of shadowing.

- The authors should elaborate more on the rationale behind (23).

- The parameter K_{max} in Table 1 is not defined in the text; moreover, I suggest to the authors to explain (or include references about) the complexity of their proposed methods.

- The selected scenario to evaluate seems quite arbitrary. I recommend justifying this selection and including at least an additional representative scenario.

- Check spelling in the legends of Figure 3 and 4.

- The conclusions are obvious and general. Deeper conclusions must be provided.

Comments on the Quality of English Language

None

Author Response

  1. Reviewer#3, Concern # 1: In the abstract, the authors explicitly refer to the GTRS acronym (which is used in the second step of the proposed method), however, they do not provide the readers with the relation between their solution and the acronym SD/SOCP, also included in the paper title.

Author response: Thank you for your careful review. We have explicitly clarified the abbreviations "SD/SOCP" and "GTRS" in the abstract and clearly explained their specific roles in the two-step method to ensure that readers can understand the composition of the method's name and its technical relevance.

  1. Reviewer#3, Concern # 2: I consider that in Section 1. Introduction, the authors should elaborate more on the scenarios or applications where node's localization is crucial.

Author response: Thank you for your valuable feedback. We have supplemented the introduction with the practical impact of insufficient positioning accuracy on wireless sensor networks (WSNs), specifically detailing its potential harm to network reliability, efficiency, and data accuracy in key applications such as environmental monitoring and autonomous systems. By citing relevant literature and providing examples, the practical significance of the research has been further highlighted.

  1. Reviewer#3, Concern # 3: In Figure 1 the "reference node" is introduced, however, it is not clearly defined in the text.

Author response: Thank you for your detailed feedback. We have explicitly defined the special role of "reference nodes" and their distinction from other anchor nodes in the system model section (Section 2) and the caption of Figure 1, ensuring a clear presentation of their function and their association with parameters such as the reference distance d_0.

  1. Reviewer#3, Concern # 4: The authors define n_i as "measurement noise", however in the context of path-loss models this parameter does not represent noise, but random attenuation resulting of shadowing.

Author response: Thank you very much for your careful review and insightful comment. You are correct that our description of "measurement noise" n_i was inaccurate. The proper characterization is that n_i represents log-normal shadowing terms, which are modeled as  n_i \sim \mathcal{N}(0, \sigma_{n_{i}}^2).

  1. Reviewer#3, Concern # 5: The authors should elaborate more on the rationale behind (23).

Author response: Thank you for your helpful suggestions. We've further substantiated the theoretical basis for the weighting scheme in formula (23) in Section 3.2. Here are the specific revisions:

“The weighting scheme is based on the fact that RSS short-range measurements are more reliable than long-range ones. The RSS measurements have a constant multiplicative factor with distance, leading to larger errors for remote links compared to nearby ones. Thus, this weight definition better reflects the reliability of different links, improving localization accuracy.”

We've added more detailed explanations about the characteristics of RSS measurements and their impact on localization accuracy. This includes:

(1) The difference in reliability between short-range and long-range RSS measurements.

(2) The effect of the constant multiplicative factor on measurement errors.

(3) How the weighting scheme effectively prioritizes more reliable measurements.

These additions should provide a clearer understanding of the rationale behind the weight definition in formula (23).

  1. Reviewer#3, Concern # 6: The parameter K_{max} in Table 1 is not defined in the text; moreover, I suggest to the authors to explain (or include references about) the complexity of their proposed methods.

Author response: Thank you for your detailed feedback. We've clearly defined the parameter k_{

max} in Section 4 (Computational Complexity Analysis) and supplemented the formulas for specific complexity calculations to support the complexity analysis of the proposed method.

  1. Reviewer#3, Concern # 7: The selected scenario to evaluate seems quite arbitrary. I recommend justifying this selection and including at least an additional representative scenario.

Author response: Thank you for your valuable suggestions. We have supplemented the basis for the selection of scenarios in Section 5 (Simulation Results).

This study selects homes and small offices (approximately 20×20 m²) as the target environments due to their widespread use and representativeness in indoor wireless localization applications. In home environments, the demand for accurate device localization is increasing, as precise position awareness enables more efficient automation and intelligent management in smart home systems. In small office settings, employee tracking can optimize resource allocation and improve overall work efficiency. Moreover, these environments have relatively well-defined boundaries and exhibit complex signal propagation characteristics, including multipath effects and signal attenuation, making them ideal for evaluating the robustness and adaptability of localization algorithms in constrained indoor spaces.

To further enhance the representativeness of the experimental scenarios, we consider two typical anchor node deployment strategies: fixed deployment and random deployment. Fixed deployment is suitable for environments with existing infrastructure, such as smart home systems or offices equipped with positioning devices, while random deployment is more relevant for dynamic, temporary, or resource-constrained applications. In this study, we choose random anchor node deployment as the primary test scenario to comprehensively evaluate the robustness of the proposed algorithm under varying anchor node distributions.

  1. Reviewer#3, Concern # 8: Check spelling in the legends of Figure 3 and 4.

Author response: Thank you for your meticulous corrections. We have thoroughly checked and rectified the spelling mistakes in the legends of Figures 3 and 4 to ensure the consistency of terms, abbreviations, and symbols.

  1. Reviewer#3, Concern # 9: The conclusions are obvious and general. Deeper conclusions must be provided.

Author response: Thank you for your valuable feedback. We've revised the conclusions section to provide more in-depth analysis and specific insights based on our study. Here are the key points we've added:

(1) Practical Implications: We've discussed the practical implications of our findings, including how the proposed method can be applied in real-world wireless sensor networks to achieve more accurate and efficient localization.

(2) Future Work: We've expanded on the potential for future research, including the extension of the method to 3D environments, integration with hybrid measurement systems, and the study of non-uniform anchor node distribution.

These revisions aim to provide deeper insights into our methodology and results, moving beyond general statements to offer specific contributions and implications of our work.

Thank you again for your helpful suggestions. We believe these revisions address your concerns and strengthen the overall quality of our paper.

Round 2

Reviewer 2 Report

Comments and Suggestions for Authors

Author improve their research paper as per my previous comment. 

Reviewer 3 Report

Comments and Suggestions for Authors

In Fig. 3 and 4 the legend says "ML-Ture", but it must be "ML-True". Please correct this mistake